# Analysis of the Efficacy and Risk Factors for Failure of Balloon Dilation for Benign Ureteral Stricture

**DOI:** 10.3390/jcm12041655

**Published:** 2023-02-19

**Authors:** Bing Wang, Wenzhi Gao, Kunlin Yang, Honglei Liu, Yangjun Han, Mingxin Diao, Chao Zuo, Minghua Zhang, Yingzhi Diao, Zhihua Li, Xinfei Li, Gang Wang, Peng Zhang, Chunji Wang, Chunjuan Xiao, Chen Huang, Yaming Gu, Xuesong Li

**Affiliations:** 1Department of Urology, Peking University First Hospital—Miyun Hospital, Beijing 100034, China; 2Department of Urology, The Third Hospital of Hebei Medical University, Shijiazhuang 050011, China; 3National Urological Cancer Center, Department of Urology, Peking University First Hospital, Institute of Urology, Peking University, Beijing 100034, China; 4Department of Urology, Emergency General Hospital, Beijing 100034, China; 5Department of Operating Room, Peking University First Hospital—Miyun Hospital, Beijing 100034, China; 6Department of Urology, Jian Gong Hospital, Beijing 100034, China

**Keywords:** balloon dilation, benign ureteral stricture, risk factor analysis, endoureterotomy

## Abstract

This study aimed to investigate the efficacy of balloon dilation in ureteral stricture and to analyze the risk factors for the failure of balloon dilation, which will hopefully provide some reference for clinicians to develop treatment plans. We retrospectively analyzed 196 patients who underwent balloon dilation between January 2012 and August 2022, 127 of whom had complete baseline and follow-up data. General clinical data, perioperative data, balloon parameters at the time of surgery, and follow-up results were collected from the patients. Univariate and multivariate logistic regression analyses were performed for the risk factors for surgical failure in patients undergoing balloon dilatation. The success rates of balloon dilatation (*n* = 30) and balloon dilatation combined with endoureterotomy (*n* = 37) for lower ureteral stricture at 3 months, 6 months, and 1 year were 81.08%, 78.38%, and 78.38% and 90%, 90%, and 86.67%, respectively. The success rates of balloon dilation at 3 months, 6 months, and 1 year in patients with recurrent upper ureteral stricture after pyeloplasty (*n* = 15) and primary treatment (*n* = 30) were 73.33%, 60%, and 53.33% and 80%, 80%, and 73.33%, respectively. The success rates of surgery at 3 months, 6 months, and 1 year for patients with recurrence of lower ureteral stricture after ureteral reimplantation or endoureterotomy (*n* = 4) and primary treatment with balloon dilatation (*n* = 34) were 75%, 75%, and 75% and 85.29%, 79.41%, and 79.41%, respectively. Multivariate analysis of the failure of balloon dilation showed that balloon circumference and multiple ureteral strictures were risk factors for balloon dilation failure (OR = 0.143, 95% CI: 0.023–0.895, *p* = 0.038; OR = 1.221, 95% CI: 1.002–1.491, *p* = 0.05). Balloon dilation combined with endoureterotomy in lower ureteral stricture had a higher success rate than balloon dilation alone. The success rate of balloon dilation in the primary treatment of the upper and lower ureter was higher than that of balloon dilation in the secondary treatment after failed repair surgery. Balloon circumference and multiple ureteral strictures are risk factors for balloon dilation failure.

## 1. Introduction

Ureteral stricture can be caused by a variety of factors, and it is mainly classified as benign or malignant. Benign ureteral stricture is the narrowing of the ureteral lumen due to non-malignant factors such as surgical injury, stones, infection, trauma, radiation therapy, kidney transplantation, and urinary diversion, while malignant ureteral stenosis can result from primary ureteral tumors or from the development of adjacent tumors [1,2]. Ureteral stricture can lead to urinary tract obstruction, hydronephrosis, and, consequently, cortical thinning and renal hypofunction.

The main treatment modalities for ureteral stricture are ureteral reconstruction surgery and endoluminal surgery [3,4]. Despite the fact that ureteral reconstruction surgery is considered the gold standard for ureteral stricture treatment, repair surgery is difficult and has many complications. Advances in endoscopic technology have provided urologists with an alternative to reconstructive surgery in the treatment of ureteral stricture [5,6,7]. Balloon dilation is one of the most common endoluminal treatment modalities for ureteral stricture [8,9,10,11,12]. Despite the fact that balloon dilation has a higher rate of stricture recurrence than reconstructive ureteral surgery, it is less invasive and has fewer complications, and its short-term treatment results are good, making it widely used in clinical practice [13,14,15]. In previous studies, the factors contributing to the recurrence of ureteral stricture after balloon dilation may be related to the patient’s gender, age, BMI, length of ureteral stricture, site of ureteral stricture, history of urologic surgery, time of surgery, balloon circumference, balloon pressure, duration of dilation, and type of postoperative ureteral stent [16]. However, there was a high degree of heterogeneity in terms of the optimal balloon circumference, balloon pressure, and duration of dilation [17].

Since there are no clear guidelines on the selection of balloon size for balloon dilation, this study further investigated the effect of different balloon sizes on balloon dilation. By investigating the risk factors for balloon dilatation failure, this study provides some reference for clinicians to develop treatment plans.

## 2. Materials and Methods

### 2.1. Study Population

A total of 196 patients with ureteral stricture underwent balloon dilation between January 2012 and August 2022 at the Peking University First Hospital, Miyun Hospital, Emergency General Hospital, and Jian Gong Hospital. Baseline data and follow-up data were complete for 127 patients, including 97 patients with balloon dilation and 30 patients with balloon dilation combined with endoureterotomy (Figure 1 and Table 1). The patient’s phone number was empty and wrong, and the patient or his family members refused to follow up and lost contact. Clinical data, including general clinical data, perioperative data, balloon parameters at the time of surgery, and follow-up results, were collected in our Reconstruction of the Urinary Tract: Epidemiology and Result (RECUTTER) database [18].

Postoperative results of ureteral stenosis patients were collected at 3 months, 6 months, and 1 year after surgery, and the results of balloon dilatation treatment were evaluated at different periods and patients in each period were divided into two groups (failure group and non-failure group). One of the following conditions resulted in patients being allocated to the failure group: the reappearance of ureteral stricture and worsening of hydronephrosis on routine imaging (intravenous pyelogram, renal scan, retrograde pyelogram, ultrasound); the need for the repositioning of the DJ tube/nephrostomy tube; or surgical treatment. The remaining patients were allocated to the non-failure group.

The risk factor analysis included the following criteria: (a) patients with ureteral strictures detected through ureteroscopy or imaging. Imaging examinations include intravenous urography, computed tomography urogram, and magnetic resonance urography; (b) treatment with balloon dilation only and with technical success. Technical success is defined as the ability of the ureteroscope and guidewire to pass through the balloon-dilated ureteral stricture and successful retention of a double J ureteral stent. Exclusion criteria: (a) contraindication to surgery; (b) combined pregnancy; (c) conditions that the investigator needed to exclude; (d) incomplete data; and (e) a combined history of malignancy. The surgical outcome 3 months postoperatively was considered as the endpoint. In an analysis of risk factors for the failure of balloon dilatation alone, data from 97 patients were considered valid. 

To compare the efficacy of balloon dilation versus balloon dilation combined with endoureterotomy in the treatment of lower segment ureteral stricture, patients with multi-segmental strictures were excluded.

The endpoint was clarified by a telephone follow-up, and the true validity of the clinical data was verified in 127 patients. The information collected included gender, age, BMI, preoperative renal function (preoperative creatinine, preoperative urea nitrogen, preoperative blood eGFR), ureteral stricture length, ureteral stricture site, whether there was a history of urologic surgery, balloon circumference, balloon pressure, and duration of dilation. The risk factors for recurrence after the balloon dilation of ureteral stricture were investigated according to whether the patients’ surgical outcome had failed or been successful, as of the endpoint.

A total of 127 patients with complete data after follow-up were included in this study. There were 97 patients with balloon dilatation alone and 30 patients with balloon dilatation combined with endoureterotomy. There were 67 patients with lower segment ureteral stricture, of which 30 received balloon dilatation combined with endoureterotomy and 37 received balloon dilatation alone. Among 97 patients who underwent balloon dilatation alone, 10 patients had multiple ureteral stenosis, 4 patients had middle ureteral stenosis, and 83 patients had upper and lower ureteral stenosis.

This study was designed and conducted in accordance with the principles of the Declaration of Helsinki (revised 2013) and was approved by the Ethics Committee of Peking University First Hospital. Individual consent was waived due to the retrospective nature of this study.

### 2.2. Operation Methods

The surgical approaches included endoscopic balloon dilatation and balloon dilatation combined with endoureterotomy (Figure 2 and Appendix A).

The main procedure for balloon dilation is to place the balloon dilator (F18–F30) along the guide wire after passing the stricture section during percutaneous nephroscopy or ureteroscopy. Then, the balloon is placed at the stricture site under direct vision, the pump is pressurized to 15–30 atm, the balloon is filled and kept filled for 3–5 min, and the balloon dilation catheter is removed. In the cases in this study, two ureteral stents were placed.

Balloon dilation combined with endoureterotomy is primarily performed by incising the end of the ureter with electrodes and then placing a balloon dilator along the guide wire.

### 2.3. Statistical Analysis

Excel Software (version 2019) was used for data management and SPSS version 22.0 was used to perform statistical analysis. Quantitative variables included age, BMI, preoperative renal function (preoperative creatinine, preoperative urea nitrogen, preoperative blood eGFR), ureteral stricture length, balloon circumference, balloon pressure, and duration of dilation, and the qualitative variables included gender, history of urologic surgery, and ureteral stricture site. Normally distributed measures were expressed as the mean ± standard deviation; skewed data were described by the median (extreme deviation). For continuous variables, the variables that were normally distributed were analyzed using the t test, and variables that did not conform to a normal distribution were analyzed using the Mann–Whitney U test. Categorical variables were analyzed using Fisher’s exact probability test. Univariate logistic regression analysis (*p* < 0.05) and multivariate logistic regression analysis (*p* < 0.1) were used for independent risk factor analysis. 

## 3. Results

Comparison of 3-month, 6-month, and 1-year outcomes of balloon dilation versus balloon dilation combined with endoureterotomy for lower ureteral stricture.

The clinical data of 67 patients who received lower ureteral stricture are shown in Table 2. The success rates of surgery for lower ureteral stricture treated with balloon dilation at 3 months, 6 months, and 1 year were 81.08%, 78.38%, and 78.38%, respectively. The success rates of balloon dilatation combined with endoureterotomy for lower ureteral stricture at 3 months, 6 months, and 1 year were 90%, 90%, and 86.67%, respectively. The age of the patients in the balloon dilation group and the combined with endoureterotomy group was 40.76 ± 14.77 and 31.27 ± 15.36, respectively, and there was a statistical difference between the two groups (*p* = 0.014). The length of ureteral stricture was 1.29 ± 0.57 (cm) and 1.86 ± 1.04 (cm) in the balloon group and combined endoureterotomy group, respectively, with a statistical difference between the two groups (*p* = 0.027).

2.Comparison of the efficacy of balloon dilatation in the primary treatment and in the secondary treatment.

The clinical data of 83 patients who received balloon dilation after primary balloon dilation and repair surgery are shown in Appendix A. The success rates of balloon dilation at 3 months, 6 months, and 1 year in patients with recurrent upper ureteral stricture after pyeloplasty and primary treatment were 73.33%, 60%, and 53.33% and 80%, 80%, and 73.33%, respectively. The success rates of surgery at 3 months, 6 months, and 1 year for patients with recurrence of lower ureteral stricture after ureteral reimplantation or endoureterotomy versus primary treatment with balloon dilatation were 75%, 75%, and 75% and 85.29%, 79.41%, and 79.41%, respectively.

3.Analysis of risk factors for failure of balloon dilatation.

The clinical data of 97 patients who underwent balloon dilation are shown in Table 3. The success rates of balloon dilation for ureteral stricture at 3 months, 6 months, and 1 year were 78.35%, 77.32%, and 71.13%. The circumference of balloon dilation was 25.83 ± 3.02 (cm) in the failed group and 24.09 ± 3.02 (cm) in the non-failure group, with a statistical difference between the two groups (*p* = 0.032). 

There was a statistical difference between the failure group and the non-failure group for the presence of ureteral multi-segmental stricture (*p* = 0.022). Univariate logistic regression and multifactor logistic regression analysis showed that balloon circumference and multi-segmental ureteral strictures were independent risk factors for the failure of balloon dilation, as shown in Table 4. Preoperative creatinine, preoperative urea nitrogen, and preoperative eGFR jointly reflected preoperative renal function, and there was no statistically significant difference between the failed and non-failure groups (*p* > 0.05).

## 4. Discussion

Ureteral stricture can lead to obstructive lesions in the urinary tract and, if left untreated or if not treated in a timely manner, can lead to hydronephrosis or even irreversible renal failure. Surgical repair is considered the gold standard in the treatment of ureteral stricture, but repair surgery is more damaging to the patient and is unable to be tolerated by some patients [19,20,21,22]. Endoluminal surgery is now also used as a primary modality for ureteral stricture treatment because it is minimally invasive and quick [23]. Balloon dilation is one of the most common endoluminal treatment modalities and has a high success rate [13,24,25,26].

Balloon dilation was first applied to treat coronary obstruction by Gruntzig et al. in 1973 [27]. With the success of the use of balloon dilation in vascular stricture, balloon dilation was gradually promoted in ureteral stricture. Lu et al. [16] performed a meta-analysis of balloon dilation for ureteral stricture and found that the combined technical success rate of endoscopic balloon dilation for benign ureteral stricture was 89% ± 4% using a random effects model, with a short-term success rate (3 months postoperatively) of 60% ± 10% and a long-term success rate (6–12 months postoperatively) of 54% ± 14%. The success rates of balloon dilation for ureteral stricture at 3 months, 6 months, and 1 year in this study were 78.35%, 77.32%, and 71.13%, respectively. Balloon dilation has a good therapeutic effect for the treatment of ureteral stricture.

In recent years, endoluminal techniques have gradually extended from balloon dilation to electrocautery, laser, and cold-knife endoureterotomy [5,14,28]. Bannakij Lojanapiwat et al. [5] found a six-month endoluminal ureteral stricture group power of 9/12 (75%) and a balloon dilation group power of 9/15 (60%). The success rate of balloon dilation in previous studies ranged from 45% to 100%, with an overall success rate of 51%, and the success rate of endoluminal endoureterotomy ranged from 62% to 95% [5]. In the present study, the success rates of balloon dilation at 3 months, 6 months, and 1 year of surgery were found to be 81.08%, 78.38%, and 78.38%, respectively, for lower ureteral stricture. The success rates of balloon dilation combined with endoureterotomy for lower ureteral stricture at 3 months, 6 months, and 1 year were 90%, 90%, and 86.67%, respectively. Balloon dilatation combined with endoureterotomy resulted in better outcomes.

Some patients failed to achieve the expected outcome after pyeloplasty for upper ureteral stricture, and the second surgery was more difficult because of severe scarring and fibrosis [29]. In this study, we found that the success rate of the primary treatment of upper ureteral stricture with balloon dilation was higher than that of the second surgery with balloon dilation after the failure of the primary pyeloplasty. Similarly, the primary treatment of lower ureteral stricture with balloon dilatation had a higher success rate than the second failed balloon dilatation after ureteral reimplantation. Patients should carefully consider their choice of primary treatment as their decision has a greater impact on their prognosis and secondary treatment.

In previous studies, preoperative renal function, length of ureteral stricture, time to onset of stricture, and whether the stricture is ischemic were the main risk factors for the failure of balloon dilation [30]. However, no agreement has yet been reached on the type of balloon, dilation pressure, and duration of dilation for the balloon dilation technique [16,31]. There are no clear criteria for the selection of balloon pressure, balloon circumference, and balloon dilation time [32,33]. Peng et al. [34] suggested that a direct application of a maximum pressure of 30 atm to ensure complete dilatation of the stricture would provide a better treatment outcome. However, in this study, it was found that balloon pressure and balloon circumference were negatively correlated with patient prognosis; choosing the maximum balloon pressure and circumference, regardless of the patient’s specific ureteral stricture, may not be the most appropriate treatment modality as too large a balloon circumference and pressure may lead to a poor prognosis.

The vast majority of patients enrolled in this study had ureteral stricture lengths of 2 cm or less, which may be the reason why ureteral stricture length was not a risk factor for balloon dilatation failure at the time of analysis [33,35]. The length of ureteral stricture found in previous studies as a risk factor for balloon dilatation failure was analyzed with a cut-off length of 2 cm [36]. The balloon dilation time was overwhelmingly 3 min, which may explain why balloon dilation time was not a risk factor for balloon dilation failure at the time of analysis.

This study also has some limitations. The failure and non-failure groups in this retrospective study were defined based on symptoms and changes in the degree of effusion, with no clear definition of renal functional status. In this study, only the balloon circumference and balloon pressure were found to be time-dependent, but they could not guide the physician to choose the appropriate balloon circumference and balloon pressure for patients with different conditions. To address this situation, we are currently designing a prospective cohort to collect data on the degree of ureteral stricture, the length of the ureteral stricture segment, the patient’s normal ureteral circumference, and the patient’s preoperative renal function, and aim to analyze the optimal balloon circumference, pressure, and dilation time for different ureteral stricture conditions.

## 5. Conclusions

In this study, the success rate of balloon dilation combined with endoureterotomy was higher than that of balloon dilation in lower ureteral stricture. The success rate of balloon dilation in the primary treatment of the upper and lower ureter was higher than that of balloon dilation in the secondary treatment after failed repair surgery. Among them, balloon circumference and multiple ureteral strictures are risk factors for balloon dilation failure. However, there is no clear protocol for patients to follow when selecting their balloon size.

## Figures and Tables

**Figure 1 jcm-12-01655-f001:**
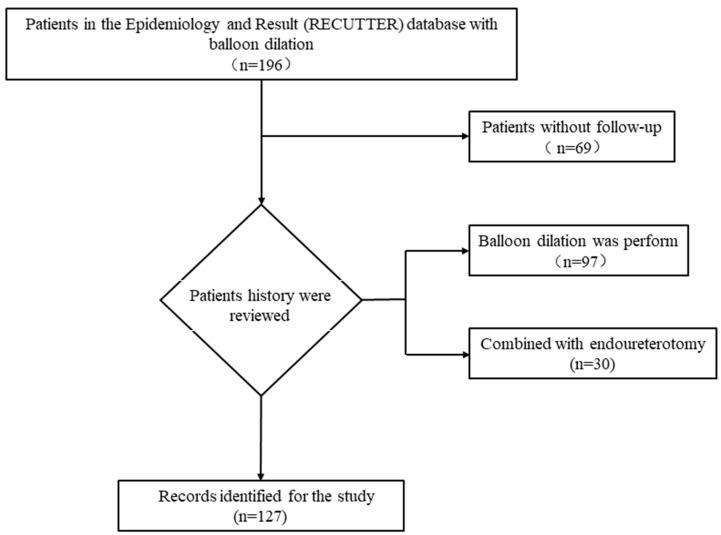
Flow chart of inclusion and exclusion criteria of the study.

**Figure 2 jcm-12-01655-f002:**
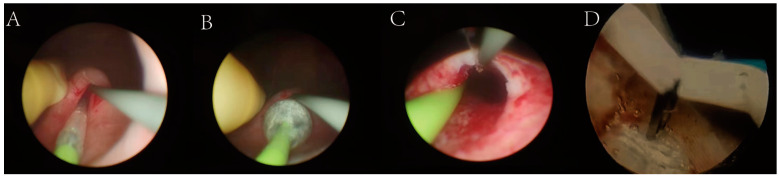
Balloon dilation and endoureterotomy. (**A**) Placement of a balloon in the stenotic segment. (**B**) The balloon is filled with water and left for 3–5 min. (**C**) The dilated ureteral stenosis was wider than before. (**D**) Cutting of stenotic segments.

**Table 1 jcm-12-01655-t001:** Patient characteristics.

Parameter	No. (%)
Patients	127
Mean age (years)	36.89 ± 14.82
BMI (kg/m^2^)	23.31 ± 3.41
Gender	
Male	58 (45.67)
Female	69 (54.33)
Treatment methods	
Balloon dilatation	97 (76.38)
Balloon dilation combined with endoureterotomy	30 (23.62)
Location of stricture	
Upper	45 (35.43)
Middle	4 (3.15)
Lower	68 (53.55)
Multiple stricture	10 (7.87)
Degree of hydronephrosis	
NA	42 (33.07)
Mild/moderate	63 (49.61)
Severe	22 (17.32)
Urethral repair surgery	
Yes	32 (25.20)
No	95 (74.80)
Preoperative SCr (μmol/L)	78.79 ± 41.70
Preoperative BUN (mmol/L)	5.14 ± 2.71
Preoperative eGFR (mL/min)	100.21 ± 25.58
Length of ureteral stricture (cm)	1.48 ± 1.19
Balloon circumference (mm)	24.85 ± 3.36
Balloon pressure (atm)	25.82 ± 4.31
Dilation time (min)	3.04 ± 4.31

**Table 2 jcm-12-01655-t002:** Baseline data for patients treated with balloon dilation versus balloon dilation combined with endoureterotomy.

Variable	Balloon Dilation	Balloon Dilation Combined with Endoureterotomy	*p* Value
Patients, *n* (%)			
Non-failure group	29 (78.38)	27 (90)	0.202
Failure group	8 (21.62)	3 (10)	
Mean age (years)	40.76 ± 14.77	31.27 ± 15.36	0.014
BMI (kg/m^2^)	22.74 ± 3.10	21.89 ± 3.54	0.324
Gender, *n* (%)			0.505
Male	14 (37.84)	9 (30)	
Female	23 (62.16)	21 (70)	
Degree of hydronephrosis			0.645
NA	9 (24.32)	13 (43.33)	
Mild/moderate	23 (62.16)	13 (43.33)	
Severe	5 (13.52)	4 (13.34)	
Preoperative SCr (μmol/L)	91.20 ± 70.76	69.87 ± 12.38	0.113
Preoperative BUN (mmol/L)	6.14 ± 6.63	4.78 ± 1.14	0.140
Preoperative eGFR (mL/min)	91.65 ± 29.88	105.69 ± 17.08	0.089
Length of ureteral stricture (cm)	1.29 ± 0.57	1.86 ± 1.04	0.027
Balloon circumference (mm)	24.21 ± 3.14	24.46 ± 3.14	0.767
Balloon pressure (atm)	26.65 ± 4.25	25.91 ± 4.23	0.537
Dilation time (min)	2.91 ± 0.63	2.79 ± 0.94	0.5617

**Table 3 jcm-12-01655-t003:** Demographic and clinical characteristics of 97 patients.

Variable	Failure Group	Non-Failure Group	*p* Value
Patients, *n* (%)	21 (21.65)	76 (78.35)	
Mean age (years)	37.62 ± 13.57	37.86 ± 15.23	0.949
BMI (kg/m^2^)	23.98 ± 2.66	23.43 ± 3.26	0.497
Gender, *n* (%)			0.238
Male	13 (61.9)	36 (47.37)	
Female	8 (38.1)	40 (52.63)	
Degree of hydronephrosis			0.696
NA	8 (38.1)	21 (21.65)	
Mild/moderate	9 (42.86)	41 (42.27)	
Severe	4 (19.04)	14 (14.43)	
Preoperative SCr (μmol/L)	74.83 ± 19.83	84.40 ± 51.86	0.422
Preoperative BUN (mmol/L)	5.09 ± 1.26	5.48 ± 3.46	0.616
Preoperative eGFR (mL/min)	101.30 ± 18.64	99.33 ± 29.48	0.798
Length of ureteral stricture (cm)	1.23 ± 0.67	1.21 ± 0.53	
Location of stricture			0.595
Upper	10 (62.5)	35 (49.3)	
Middle	0 (0)	4 (5.63)	
Lower	6 (37.5)	32 (45.07)	
Urethral repair surgery			0.303
Yes	7 (33.33)	17 (22.37)	
No	14 (66.67)	59 (77.63)	
Multiple stricture			0.022
Presence	5 (23.81)	5 (6.58)	
Absence	16 (76.19)	71 (93.42)	
Balloon circumference (mm)	25.83 ± 3.02	24.09 ± 3.02	0.032
Balloon pressure (atm)	27.25 ± 4.19	25.20 ± 4.50	0.076
Dilation time (min)	3.1 ± 0.77	3.04 ± 0.60	0.728

**Table 4 jcm-12-01655-t004:** Univariate and multivariate logistic regression analysis for failure of balloon dilatation.

Variable	Univariate			Multivariate		
	OR	95% CI	*p* Value	OR	95% CI	*p* Value
Mean age (years)	0.999	0.967–1.032	0.948			
BMI (kg/m^2^)	1.06	0.898–1.251	0.492			
Gender, *n* (%)	1.806	0.671–4.855	0.242			
Degree of hydronephrosis	1.429	0.363–5.627	0.61			
Preoperative SCr (μmol/L)	0.992	0.972–1.012	0.422			
Preoperative BUN (mmol/L)	0.947	0.763–1.175	0.619			
Preoperative eGFR (mL/min)	1.003	0.982–1.024	0.795			
Length of ureteral stricture (cm)	1.059	0.363–3.090	0.917			
Upper stricture	1.138	0.429–3.017	0.795			
Middle/lower stricture	1.8	0.654–4.956	0.255			
Urethral repair surgery	0.576	0.201–1.656	0.306			
Multiple stricture	0.225	0.058–0.872	0.031	0.143	0.023–0.895	0.038
Balloon circumference (mm)	1.193	1.008–1.413	0.04	1.221	1.002–1.491	0.05
Balloon pressure (ATM)	1.129	0.985–1.294	0.082	1.145	0.972–1.348	0.106
Dilation time (min)	1.148	0.533–2.471	0.724			

## Data Availability

Not applicable.

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
