# Peer review of "Analysis of the Efficacy and Risk Factors for Failure of Balloon Dilation for Benign Ureteral Stricture"

_jcm, 2023, doi:10.3390/jcm12041655_

Round 1

Reviewer 1 Report

Dear Editor,

I have read with interest the paper” Analysis of the efficacy and risk factors of balloon dilation for benign ureteral stricture” by Bing Wnag and collegues.

They  retrospectively analyzed 196 patients who underwent 32 balloon dilation between January 2012 and August 2022, 127 of whom had complete baseline and 33 follow-up data.

They concluded that

 The success rate of balloon dilation in the primary treatment of upper and lower ureter was higher than that of balloon dilation in the second treatment 49 after failed repair surgery and Balloon circumference and multiple ureteral strictures are risk factors for  balloon dilation failure.

The topic of this paper is well known by the urologists as ureteral stricture is a “challenge “ for clinicians.

For this reason this paper may have some “appeal” for the readers.

Having said that this manuscript ha some limitations:

-retrospective design

-low number of involved pts

-hudge heterogeneity of patients

Moreover:

-Title: risk factor of what? Probably “Analysis of the efficacy and risk factors for failure of balloon dilation for benign ureteral stricture”

-Abstract: number of patients considered for each group should be added (not only % of success)

Introduction:  adequate

Mat and Met. Authors defined 1 y of follow up a”long term follow up”. Please rephrase this sentence.

Conditions for defining recurrence are clear in my opinion… conditions for defining risk factors are less clear. Do the Authors try to do better? [The risk factor analysis included the following criteria: ….] I thik that this paraghraph is somewhat unclear.

How did the authors diagnose stenosis (what is the preop imaging for all cases) and measure it? How did the Author chose balloon or ballon + endoureteretomy?

In the abstract the Authors mentioned the reuslt of dilation for upper UT. I did not find these results, clearly, in the manuscript.

Results: lenght of stricture in B and B+E was different. In light of this, are the Authors sure that statistical analysis

Discussion: adequate

Table. Table 1. Baseline data for patients treated intrathecally. Intrathecally?

The paper needs A table with base line conditions of patients

Fig.1 Is is a orifice stenosis? I think that this kind of stenosis is different that ureteral stenosis (ie after surgery…)

Reviewer 2 Report

The authors should be congratulated for the work and for addressing an important topic. However, some points warrant mentions:

MAJOR COMMENTS:

1.    in the “Introduction” section, lines 76-79, the authors should avoid results and conclusions in this section of the manuscript.

2.    In the “Material and Methods” section, the authors should be as complete as possible, avoiding abbreviations and “etc.” as in line 92. Thus, a complete list of exclusion criteria is required.

3.    In the “Material and Methods” section, lines “118-120”, the sentence is meaningless in this position. Is it an aim? A method? I suggest to clarify.

4.    In the “Material and Methods” section, line 121, what the authors mean with “The endpoint”? it has not been previously defined. Moreover, a few lines before they declared the selection of 97 patients, whilst now are 127. Thus, the whole section is really confusing.

5.    Why did the authors consider two different p-values thresholds for univariable and multivariable logistic analysis?

6.    In the “Results” section, the number of patients went from 97 to 67 in line 159. In the whole section, the authors should better explain the numerosity of the studied population.

MINOR COMMENTS:

1.    in the “Introduction” section, the authors listed risk factors for benign strictures, however, I suggest to cite also malignant causes.

2.    In the “Material and Methods” section, the authors listed “lumbar pain” as a factor to define the failure of the procedure. I suggest to clarify the type of lumbar pain considered, such as “lumbar pain due to renal colic”.

3.    An overall revision of the English language style is required.

4.    In the “Discussion” section, lines 200 and 202 require citations.

5.    In the “Conclusions” section, I suggest to better write the first sentence. It is like a comparison between lower ureteral stricture and balloon dilation as it is.

Round 2

Reviewer 1 Report

The Authors improved the paper significantly.

Author Response

Thank you for your opinion and have a good life.

Reviewer 2 Report

Minor comments:

1 - line 88, I suggest to eliminate "lumbar pain", which does not comprehensively describe renal colic.

2 - in line 101 "The final data of 97 patients were considered valid", in line 114 "A total of 127 patients with complete data after follow-up were included in this study". I ask again to the authors to clarify this data.

3 -  I suggest to avoid "aims" in the "material and methods" section.

4 - why in the "Conclusions" section, the authors reported "intraperitoneal" urethrotomy?
